# In Situ LSPR Sensing of Secreted Insulin in Organ-on-Chip

**DOI:** 10.3390/bios11050138

**Published:** 2021-04-28

**Authors:** María A. Ortega, Júlia Rodríguez-Comas, Ozlem Yavas, Ferran Velasco-Mallorquí, Jordina Balaguer-Trias, Victor Parra, Anna Novials, Joan M. Servitja, Romain Quidant, Javier Ramón-Azcón

**Affiliations:** 1Biosensors for Bioengineering Group, Institute for Bioengineering of Catalonia (IBEC), The Barcelona Institute of Science and Technology (BIST), Baldiri I Reixac, 10-12, 08028 Barcelona, Spain; mortega@ibecbarcelona.eu (M.A.O.); jrodriguezc@ibecbarcelona.eu (J.R.-C.); fvelasco@ibecbarcelona.eu (F.V.-M.); jbalaguer@ibecbarcelona.eu (J.B.-T.); victor.a.parramonreal@gmail.com (V.P.); 2Plasmon Nano-Optics Group, ICFO-Institute for Photonics Sciences, The Barcelona Institute of Science and Technology, 08860 Barcelona, Spain; oezlem.yavas@gmail.com (O.Y.); rquidant@ethz.ch (R.Q.); 3Diabetes and Obesity Research Laboratory, Institut d’Investigacions Biomèdiques August Pi i Sunyer (IDIBAPS), 08036 Barcelona, Spain; anovials@clinic.cat (A.N.); servitja@clinic.cat (J.M.S.); 4Centro de Investigación Biomédica en Red de Diabetes y Enfermedades Metabólicas (CIBERDEM), 28029 Madrid, Spain; 5Nanophotonic Systems Laboratory, Department of Mechanical and Process Engineering, ETH Zurich, 8092 Zurich, Switzerland; 6ICREA-Institució Catalana de Recerca i Estudis Avançats, 08010 Barcelona, Spain

**Keywords:** LSPR sensors, organ-on-a-chip, in situ insulin monitoring

## Abstract

Organ-on-a-chip (OOC) devices offer new approaches for metabolic disease modeling and drug discovery by providing biologically relevant models of tissues and organs in vitro with a high degree of control over experimental variables for high-content screening applications. Yet, to fully exploit the potential of these platforms, there is a need to interface them with integrated non-labeled sensing modules, capable of monitoring, in situ, their biochemical response to external stimuli, such as stress or drugs. In order to meet this need, we aim here to develop an integrated technology based on coupling a localized surface plasmon resonance (LSPR) sensing module to an OOC device to monitor the insulin in situ secretion in pancreatic islets, a key physiological event that is usually perturbed in metabolic diseases such as type 2 diabetes (T2D). As a proof of concept, we developed a biomimetic islet-on-a-chip (IOC) device composed of mouse pancreatic islets hosted in a cellulose-based scaffold as a novel approach. The IOC was interfaced with a state-of-the-art on-chip LSPR sensing platform to monitor the in situ insulin secretion. The developed platform offers a powerful tool to enable the in situ response study of microtissues to external stimuli for applications such as a drug-screening platform for human models, bypassing animal testing.

## 1. Introduction

Type 2 diabetes (T2D) is one of the most common metabolic diseases, affecting millions of people worldwide [1]. Patients with T2D present a progressive decline in pancreatic β-cell function, mainly characterized by impaired insulin secretion. For this reason, the study of insulin secretion aimed at addressing islet functionality requires the ability to monitor insulin in situ over time, and measurements of insulin secretion dynamics are of significant clinical relevance. Traditionally, pancreatic β-cell function is assessed by measuring the insulin released by glucose-stimulated insulin secretion (GSIS) assays. These experiments involve manual liquid handling, static incubation of the islets, and enzyme-linked immunosorbent assays (ELISA) that require a long processing time.

Several approaches have emerged for engineering biomimetic, easy-to-use, and compatible organ-on-a-chip (OOC) microfluidic devices capable of reproducing physiological cell responses in vitro. Indeed, numerous micro-scale engineering OOC have been fabricated, modeling different tissues (e.g., muscle [2], blood vessels [3], liver [4], gut [5], or pancreatic islets [6]). Recent advances in miniaturizing microfluidic systems and advanced tissue fabrication procedures have enabled researchers to create multiple tissues-on-a-chip with a high degree of control over experimental variables for high-content screening applications [7,8,9,10,11].

Currently, there is a gap in the integration of these potential platforms to sensing modules, capable of monitoring in situ fast metabolic behaviors subjected to external stimuli, such as stress or drugs. Extensive efforts have been made to integrate three-dimensional (3D) tissue platforms with a sensing system for in situ continuous measurements of relevant targets [2,12,13,14]. However, the integration and application of sensing strategies are still far from providing a high throughput and reliable data to reveal the status and dynamics of the OOC.

Regarding pancreatic islets, there are only few examples where microfluidic systems have been integrated with free-labeled sensing platforms to study the dynamic of the insulin secretion profile. These works are focused on the monitoring of electrophysiology phenomena using complex microelectronic arrays with fluidic systems [15,16]. However, in those studies, neither do the biological models represent the islets in a 3D environment (biomimicking native pancreas configuration), nor can the electrochemical sensors efficiently monitor in a label-free way the secretion of insulin, as they only provide a recording of the cell activity. To fully exploit the potential of these platforms, there is a need to interface them with an integrated sensing module capable of directly monitoring the islet insulin response.

Among the different existing transduction methods, optical biosensors have the advantage of being highly sensitive, enabling label-free, cost-effective, and real-time sensing. As a well-studied optical sensing scheme, localized surface plasmon resonance (LSPR)-based sensors, which exploit the unique properties of noble metal nanostructures, have shown a great ability to detect all kinds of molecular biomarkers (proteins [17], peptides [18], mRNA [19], DNA [20,21], and miRNA [22]) in biological samples. The ease of optical transduction and the compact nature of LSPR sensors means their integration into fully automated microfluidic devices to perform multiplexed quantitative detection can be achieved [23].

In this work, we present an integrated on-chip insulin secretion study platform, combining novel islet-on-a-chip (IOC) technology interfaced with an on-chip LSPR biosensing platform (Figure 1). Unlike other IOC devices that are based on multiple tiny wells to trap the islets [24,25,26,27], which can promote shear stress-induced cell damage, we have developed an IOC that houses primary mouse pancreatic islets embedded in a non-biodegradable cellulose-based scaffold that intends to biomimic the native pancreas host. The integration of both platforms allows, for the first time, a highly sensitive and label-free monitoring of in situ insulin secretion by pancreatic islets subjected to different glucose concentrations, under physiological conditions, offering a powerful tool for future biomedicine and pharmaceutical research related to diabetes.

## 2. Materials and Methods

### 2.1. Carboxymethyl Cellulose (CMC)-Cryogel Fabrication

Carboxymethyl cellulose (CMC, 419273, Merck Life, Darmstadt, Germany) is dissolved in MilliQ water (DI) to the desired concentration of 0.5% and crosslinking initiated by adding 50 mg mL^−1^ of adipic acid dihydrazide (AAD, ref A0638, Merck Life, Darmstadt, Germany), 1 μg μL^−1^ of N-(3-Dimethylaminopropyl)-N’-ethylcarbodiimide hydrochloride (EDC, E7750, Merck Life, Darmstadt, Germany), and MES buffer 0.5 M, pH 5.5. To stain the CMC cryogels, aminofluorescein (Merck Life, Darmstadt, Germany) was added to the prepolymeric solution in case the fibers need to be stained. The reaction mixture is rapidly dispensed inside a mold and placed overnight at −20 °C resulting in ice crystal nucleation. Finally, the cryogels are thawed and washed consecutively by submerging them in DI, 100 mM NaOH (Panreac, Darmstadt, Germany), 10 mM ethylenediaminetetraacetic acid (EDTA, 03690, Merck Life, Darmstadt, Germany), and 3 times in PBS. Once finished, the cryogels were autoclaved for further experiments.

### 2.2. Characterization of CMC Cryogels

The swelling ratio indicates, quantitatively, the water uptake capability of the scaffold. After the cryogel fabrication, scaffolds were dried at room temperature for 2 days and weighted. Subsequently, the cryogel was submerged into MilliQ water for 4 days until it reached equilibrium state and was weighted for a second time. For the swelling measurements, Equation (1) was used:Swelling ratio = (*Weq* − *Wd*)/*Weq* × 100(1)
where *Weq* represent the scaffold equilibrium weight and *Wd* is the scaffold dry weight. A total of 3 cryogels per condition were measured in this assay. On the other side, stiffness measurements were obtained from compression assays using a Zwick Z0.5 TN instrument (Zwick-Roell, Ull, Germany) with 5 N load cell. Compression assays were performed with samples at room temperature up to 30% final compression range at 0.1 mN of preloading force and at 20%/min of strain rate. Finally, the Young’s modulus was calculated from the slope of the curve in a range from 10% to 20% of compression.

Scanning electron microscopy (SEM) characterization was performed using a NOVA NanoSEM 230 at 10 kV. Different washing steps were performed using ethanol as a solvent, gradually incrementing its concentration from 50% to 99.5%. Samples were treated with critical point drying and carbon sputtering before the SEM acquisition.

### 2.3. Mouse Pancreatic Islet Isolation

Mouse pancreatic islets were isolated from 8- to 10-week-old C57BL/6J male mice by collagenase (Roche, Basel, Switzerland) digestion of the pancreas followed by Histopaque gradient (Sigma-Aldrich, St. Louis, MO, USA), as described previously [28]. Islets were cultured for 24 h at 37 °C and 5% CO_2_ in RPMI 1640 medium (11.1 mM glucose) supplemented with 10% FBS (*v/v*), 2 mM glutamine, 100 units/mL penicillin, and 100 μg mL^−1^ streptomycin before performing the experiments. Experimental procedures were approved by the Animal Ethics Committee of the University of Barcelona according to the Principles of Laboratory Animal Care.

### 2.4. Gene Expression Analysis

The miRNeasy kit (ref 74204, Qiagen, Hilden, Germany) was used to extract total RNA, and the high-capacity cDNA reverse transcription kit (ref 4368813, ThermoFisher Scientific, Carlsbad, CA, USA) was used to reverse transcribe it. Gene expression was examined by quantitative Polymerase Chain Reaction (PCR) using SYBR Green (ref 1178401K, Invitrogen, Carlsbad, CA, USA) in a 7900HT Fast Real-Time PCR System (ref 4329001, Applied Biosystems, Foster City, CA, USA). The primer sequences used are listed in Table 1. The expression levels of genes of interest were normalized to the expression of Tbp1.

### 2.5. Glucose-Stimulated Insulin Secretion (GSIS)

Islets housed within CMC cryogels were transferred into the microfluidic chip and were allowed to settle to the bottom of the chamber for 24 h. Subsequently, they were preincubated with Krebs–Ringer bicarbonate HEPES (KRBH) buffer solution (115 mM NaCl, 24 mM NaHCO_3_, 5 mM KCl, 1 mM MgCl_2_·6H_2_O, 1 mM CaCl_2_·2H_2_O, and 20 mM HEPES, pH 7.4) containing 11.1 mM glucose for 30 min at 37 °C (basal condition). The cryogels were then incubated at 2.8 mM glucose, followed by perfusion with KRBH solution with 16.7 mM glucose. First, supernatants were collected, and the cellular insulin contents were recovered in an acid-ethanol solution. Insulin concentration was determined by Insulin Mouse ELISA. For in situ and label-free detection of insulin levels, we integrated the microfluidic chip with the on-chip LSPR platform.

### 2.6. Immunofluorescence

Cryogels stained with aminofluorescein (green) were fixed with 10% formalin solution (Merck Life, Darmstadt, Germany) for 30 min and were then permeabilized with 0.5% Triton X-100 (Merck Life, Dorset, UK) and blocked by adding 3% donkey serum (Merck Life, Darmstadt, Germany). The cryogels were incubated overnight at 4 °C with the primary antibody anti-insulin (mouse anti-insulin (+proinsulin) monoclonal antibody 1:500; ref BM508, OriGene EU, Herford, Germany) to stain the insulin from the pancreatic β-cells. Subsequently, secondary antibody was added for 2 h at room temperature (ref A32, AlexaFluor 555 conjugate anti-mouse 1:250; Life Technologies, Carlsbad, CA, USA). 4′,6-diamidino-2-phenylindole (DAPI) (1:1000; ThermoFisher Scientific, Carlsbad, CA, USA) was used to counterstain the nuclei. Fluorescent images were obtained using confocal microscopy (LSM 800 microscope model, Zeiss, Orberkochen, Germany).

### 2.7. Immunoreagents and ELISA Immunoassay Protocol

The 96-well plate (Polystyrene Maxisorp 96 well microplates, Nunc, Roskilde, Denmark) was coated with 50 µL per well of the capture antibody (also used as the capture antibody in the LSPR measurements) mouse anti-insulin monoclonal antibody (ref NB100-73008, clone 3A6, Novus biologicals, Littleton, CO, USA) at 4 µg mL^−1^ prepared in a coating buffer (0.05 M of Na_2_CO_3_/NaHCO_3_, pH 9.6). The plate was washed and 8 solutions of recombinant human insulin (ref 91077C, Merck Life, Darmstadt, Germany) from 580 to 0 ng mL^−1^ prepared in PBST (PBS = 0.01 M phosphate buffer, 0.14 M NaCl, and 0.003 M KCl, with 0.05% (*v/v*) Tween 20 at pH 7.5) was added as an internal calibration curve together with the samples to interrogate (50 µL/well). The plate was incubated at room temperature for 1 h. A second wash step was performed, and detection antibody (Biotinylated Insulin Antibody (ref NB100-64697B, clone D3E7 (5B6/6), Novus Biologicals, Littleton, CO, USA) prepared in PBST at 0.031 µg mL^−1^ was added (50 µL per well) and incubated at RT for 30 min. Finally, 50 µL/well of streptavidin-horseradish peroxidase (SAv-HRP) solution at 0.25 µg mL^−1^ prepared in PBST was added and incubated for 30 min at RT. Following that step, 50 µL/well of the substrate solution was added and incubated for 3–5 min, protected from light. Finally, 50 µL/well of H_2_SO_4_ 4 N was added to stop the enzymatic reaction. The absorbances were read at 450 nm. Calibration curves were fitted using a sigmoidal fit function.

### 2.8. Fabrication of IOC Microfluidic Platform

The microfluidic chip was firstly designed using CleWin software and fabricated using a standard soft lithography replica molding technique. Briefly, a silicon wafer mold was created through a one-layer process using negative photoresist SU8-2100 (MicroChem, Westborough, MA, USA). The microfluidic chip design was printed on a high-quality acetate film to be used as a mask, and finally a microfeatured master mold was then obtained by contact photolithography. To obtain a polydimethylsiloxane (PDMS) fluidic chip, a mixture of prepolymer with curing agent (Sylgard 184, Dow corning, Midland, TX, USA) was prepared at a 10:1 ratio, degassed in a vacuum chamber for 1 h, and poured on the SU8 master mold. The polymer mix quantity was calculated to obtain a 3 mm layer (Layer 2, see Figure 4(ai)). After 4 h at 80 °C in an oven, the PDMS replica was cured and carefully peeled off from the mold. Holes were punched both for the entry and exit of liquids. In parallel, a 2 mm layer of PDMS (Layer 1, see Figure 4(aii)) was prepared (using a non-patterned silicon mold), cured, and cut out. The two layers were finally bonded irreversibly by oxygen plasma activation (Expanded Plasma Cleaner, PCD-002-CE Model, Harrick Scientific Corporation, Ossining, NY, USA), and chambers for the CMC-islet scaffolds were punched. The final microfluidic chip was bound to a standard cover slide, allowing handling and visualization under the microscope if needed (Figure 4(aiii)). Finally, a customized glass cover (37 mm × 20 mm) was activated using oxygen plasma and bound to the PDMS chip irreversibly in order to seal the chip (Figure 4(aiv)).

### 2.9. Statistics

Statistical analysis was performed using Graph Prism software (GraphPad Software, San Diego, CA, USA). Data are expressed as the mean ± SEM, and statistical significance was determined by two-tailed Student’s *t*-test. Results were considered significant at *p* < 0.05.

## 3. Results and Discussions

### 3.1. Fabrication of a 3D Effective Cellulose Matrix to House Pancreatic Islets

We developed a functional islet-on-a-chip (IOC) microfluidic device to monitor insulin secretion under flow conditions. It is known that platforms of islet perfusion mimic in vivo physiology better than static culture systems, therefore improving the islet health [29]. Islets of Langerhans are clusters of cells within the pancreas that are responsible for the production and secretion of different hormones that regulate circulating glucose levels. β-cells are the predominant cell type within the pancreatic islets in mammals and the unique source of circulating insulin, being fundamental for the maintenance of glucose homeostasis [30,31,32]. Unlike the other IOC devices that are based on multiple tiny wells to trap the islets [24,25,26,27,33,34,35,36], we have precisely engineered a heterogeneous porous cryogel scaffold which offers a robust approach for spatially organizing the islets, and which can limit shear-induced cell damage. It was recently demonstrated that 3D polymeric-based scaffolds offer mechanical and chemical properties that make them valuable in tissue engineering applications [37].

The most extensively utilized technique to achieve in vitro tissue engineering is to use the encapsulated hydrogels which present a high-water content and highly resembling in vivo physical properties [38,39,40,41,42]. However, conventional hydrogels present several limitations due to the small pore size. They present an inadequate diffusion of oxygen and nutrients/waste products, as well as limited cellular mobility and cell spreading. To address these challenges, we used the cryogelation technique, a procedure that allows the formation of cryogels at sub-zero temperatures. Typically, the liquid prepolymer solution is cooled at −20 °C. At this temperature, a large percentage of the material crystallizes due to its water content. When thawed, the ice crystals leave behind empty spaces, allowing us to obtain different pore diameters, as shown in Figure 2a. Following this principle, we can generate a 3D extracellular matrix mimicking scaffolds with a specific range of porosity (Figure 2b,c), in which the islets can be seeded, allowing the transfer of oxygen, nutrient and waste removal, and avoiding possible apoptosis or cell death. The scaffold properties can be modulated simply by altering the concentrations of the polymer and varying the freezing temperature [43]. As the cryogel technique allows us to achieve a micro-range porosity with a wide distribution range, and mouse pancreatic islets are diverse in size (~50–150 µm in diameter) with an average size of 100 µm, we determined 0.5% of CMC cryogel as a potential suitable niche for the islets (Figure 2b,c). The designed carboxymethyl cellulose (CMC) cryogel presents several advantages—besides its high porosity, it also offers the mechanical strength required for housing pancreatic islets, with a stiffness of 0.67 ± 0.1 KPa and a swelling ratio of 98.1% ± 0.3% [44] (Figure 2d), as well as being elastic. It is a non-degradable material from mammalian cells and it also allows surgical sterility by means of autoclaving [45]. Indeed, we have recently demonstrated that CMC scaffolds can be used to generate functional pseudoislets from insulin-producing INS1E-cells, representing a suitable technique to generate β-cell clusters and to study pancreatic islets in vitro [46].

Pancreatic islets were obtained from C57BL6 wild-type mice as described elsewhere [47]. A total of 30 islets were seeded in a 0.5% CMC cryogel as shown in Figure 3a and were allowed to recover overnight prior to performing the microfluidic experiments. Figure 3b shows the bright field image of the pancreatic islets housed in a CMC scaffold and immunofluorescent confocal images of the islets integrated within the cellulose fibers. The gene expression analysis of the three β-cell-specific transcriptional regulators and positive indicators of β-cell health and functionality, *Pdx1* (pancreas/duodenum homeobox protein 1), *MafA* (V-maf musculoaponeurotic fibrosarcoma oncogene homolog A), and *NeuroD1* (Neuronal Differentiation 1), revealed no significant differences when comparing the islets housed within CMC-based scaffolds and isolated pancreatic islets in suspension (Figure 3c). Additionally, the stress markers *Chop* (C/EBP homologous protein), *Trib3* (Tribbles pseudokinase 3), and *Atf3* (activating transcription factor 3) did not present significant differences either (Figure 3c), indicating that our cellulose-based cryogel provides a physiologically relevant environment and facilitates the diffusion of oxygen and nutrients, as well as demonstrating that islets do not suffer stress when integrated inside the CMC scaffold.

### 3.2. Islet-on-a-Chip Microfluidic Platform

A microfluidic device was designed and fabricated to host the in vitro model integrated by the CMC cryogel and mouse pancreatic islets. The dimensions of the device are shown in Figure 4a. Two circular chambers with a diameter of 10 mm were designed, where the CMC islets 3D in vitro model is located. Microfluidic channels of 1 mm width and 0.20 mm height were designed to connect those chambers and enable the circulation of the liquids inside the device (Appendix A).

The microfluidic IOC device was fabricated using a standard soft lithography replica molding technique as previously described in the methods section. The IOC microfluidic chip is integrated by two layers of PDMS with the purpose of elevating the microfluidic channels and creating a pool where the scaffold with the islets can be located, decreasing possible shear stress produced by a direct flow (Figure 4(ai,aii)). The device provides biomimicking of the physiological environment of the organ, supplying nutrient and oxygen exchange to the 3D in vitro construct. The flow profile inside the microfluidic device was simulated by COMSOL Multiphysics Software. The fluid velocity field for each intersection was solved using the laminar flow physics module with a customized mesh (3327 tetrahedral elements). The boundary conditions of the inlets were defined by the channel geometry, resistance of 4.8 × 10^10^ Pa s m^−3^, operational flow rate of 50 µL min^−1^, and an inlet pressure of 39.84 Pa. The remaining boundaries were specified as walls (no-slip boundary condition) and the material filling the channels was chosen as water under an incompressible flow. A stationary solver was used for the calculations. In order to create a realistic approximation, a solid cylinder with the dimensions and mechanical and chemical properties of the CMC scaffold was incorporated in the simulations. Appendix A show the velocity profile inside the whole device. The red zones indicate a higher flow velocity (7 × 10^−4^ m s^−1^ at the well mouth and 3.69 × 10^−2^ m s^−1^, as a maximum velocity, in the center of the chamber) appearing in the boundaries of the scaffold. The study shows that the flow rates and geometry used during the experiments do not affect the stability of the 3D construct and, additionally, do not exert shear stress to the cell system as a consequence of flow, showing a maximum velocity of 0.03 mm s^−1^ (red zones in Figure 4b). Figure 4c shows a real picture of the microfluidic device with the CMC islets fabricated inside the chamber. The fluidic system also helps the delivery of the secreted insulin from the IOC to the on-chip LSPR sensing platform.

We examined the functionality of the islets housed within the CMC cryogel before running microfluidic measurements by performing a glucose-stimulated insulin secretion (GSIS) assay without flow inside the IOC device. Initially, the islets were incubated with 2.8 mM glucose, a condition that dampens the secretory capacity of the β-cells, followed by incubation with 16.7 mM glucose. Our results show that the islets remain as functional units in the cryogel scaffolds, validated by the release of insulin from the pancreatic β-cells in response to glucose and quantified by a conventional ELISA (enzyme-linked immunosorbent assay). Figure 4d shows how a high glucose concentration (16.7 mM) causes a time-dependent insulin secretion from the β-cells, normalized by the total insulin content of the islets. The basal condition represents the accumulated insulin released by the islets 30 min after an overnight culture in media containing 11.1 mM glucose. Having demonstrated that islets housed within the cryogel respond to glucose, we set up the integration of the LSPR system into the microfluidic system in order to detect the insulin levels in situ.

### 3.3. On-Chip LSPR Measurements

The on-chip LSPR platform was incorporated to quantify the insulin levels from the IOC device. The platform is a state-of-the-art integrated opto-fluidic module that had been previously used for the detection of several protein biomarkers [17,23]. It enables parallel and controlled measurements on a single chip (2.5 × 2.5 cm^2^), reducing the reagent volumes, and providing in situ and label-free detection of insulin concentrations in the samples. The LSPR sensing regions consist of arrays of gold nanorods, fabricated by electron beam lithography on a glass substrate, using the optimized parameters of references [17,23,48]. The LSPR peak was set to be around 800 nm as measured by a custom-built transmission microscopy set-up integrated with a spectrometer. The optical set-up uses a galvanometric mirror to interrogate up to 32 different sensing regions in parallel. Our data analysis software delivers both peak and centroid positions of the sensed regions in situ [23]. To complete the assembly of the LSPR sensing platform, the gold sensors on glass were integrated into a microfluidic environment (Figure 5a). The latter, built by multilayer soft lithography [49], consists of two layers of PDMS networks, namely, flow and control layers. The flow network, hosting the LSPR sensors, is where the insulin detection measurements are performed. In the upper layer, the control network includes pneumatic “Quake valves” that are used to control the reagent flow through the underneath channels. Each valve is individually managed by an external controller that enables full automation of the successive steps of the detection bioassay. The microfluidic architecture includes parallel channels that are individually addressable to perform, on the same chip, eight parallel measurements with up to four replicas, which can be easily used for multiplexed experiments.

The Au sensors were biofunctionalized by immobilizing the insulin antibody to capture the insulin from the sample. To this end, a self-assembled monolayer of MUA (mercaptoundecanoic acid) was formed on the nanorods prior to chip assembly. The assembled chip with a stable MUA layer was then used to immobilize a monoclonal antibody against insulin through EDC/NHS chemistry. A scheme with the functionalization strategy is shown in Figure 5b. Once the antibody is immobilized on the sensors from all eight parallel channels, the separate channels are then used to detect, consecutively, the secreted insulin from the IOC device at eight different times. The eight channels chip design allowed us to divide the whole sensor array in individual sensing areas, monitoring the secreted insulin from the IOC device in situ and in a continuous way. Prior to running the sample measurements, we optimized the antibody concentration (Figure 5c) obtaining a saturation plateau at 100 µg mL^−1^. Sensograms are shown in Appendix A corresponding to 2 h of functionalization of 0, 10, 50, and 100 µg mL^−1^, and a functionalization of 200 µg mL^−1^ (which reached the saturation in a shorter time ~45 min), respectively. Furthermore, a direct detection of a recombinant insulin solution of 5 µg mL^−1^ prepared in basal (11.1 mM), low (2.8 mM), and high (16.7 mM) glucose conditions were compared to study the bulk refractive index effect on LSPR measurements. Appendix A reveals that there is no significant bulk refractive index effect observed for the glucose concentrations considered here, greatly simplifying the integration of the on-chip LSPR platform with the IOC device. All experiments presented here were performed under direct detection mode (without the need for a secondary antibody to amplify the signal), which allows the continuous monitorization of secreted insulin from a connected IOC device.

For the reference of insulin detection of the IOC samples, the eight-point calibration curve of insulin in KRBH buffer with 16.7 mM glucose in eight parallel channels was obtained. The calibration curve (Figure 5d) shows a limit of detection (LOD) of 0.85 ± 0.13 µg mL^−1^ and EC50 of 5.6 ± 1.2 µg mL^−1^. Every data point represents the mean value of three on-chip replicas, and error bars stand for the standard deviations. The IOC device with CMC islets was connected to the LSPR sensing platform to interrogate the secreted insulin. Basal, low (2.8 mM), and high (16.7 mM) glucose levels in KRBH buffer were used to stimulate the CMC islets in the IOC device for different durations in fluidic conditions. Every 30 min was defined as a cycle, and every cycle was flown into separate channels of the LSPR chip for the detection of the accumulative secreted insulin concentration from the IOC device during that cycle (Figure 5e). The raw data obtained for the LSPR measurements during these measurements are shown in Appendix A. The insulin secretion profile obtained by LSPR measurements shown in Figure 5e reveals that our integrated platform was able to detect an incremental insulin secretion by the pancreatic islets in response to high glucose stimulation over time. The basal level corresponds to insulin accumulated for a time interval of 30 min inside the chamber. With the aim to validate these results, a second IOC device under the same experimental conditions was implemented and the samples were interrogated by ELISA technique. Even though the results cannot be directly compared between both techniques, the insulin secretion profile shown in Appendix A reveals the same trend in insulin secretion: a remarkable increment of the insulin levels in response to a high glucose content, which supports the results obtained by the on-chip LSPR platform. These results are the preliminary steps to monitor, in a continuous way, the dynamics of insulin secretion by native pancreatic islets under physiological conditions. They demonstrate the potential of the integrated platform to perform OOC experiments with real-time insulin detection, providing a strong tool for drug testing, toxicity studies, and the elucidation of secretion dynamics in relevant tissues linked to metabolic diseases, such as T2D.

## 4. Conclusions

In this work, we present an OOC platform integrated with an on-chip biosensing platform, enabling in situ monitoring of the insulin secretion from an OOC device. Previously described microfluidic perfusion systems aimed at studying islet functionality are based on the off-line quantification of insulin by ELISA or on-line insulin detection by means of immunofluorescence, therefore labeling this hormone with antibodies. To our knowledge, this is the first time that an IOC device has been coupled with an LSPR sensing module to monitor, in situ and label-free, the insulin released by pancreatic islets. This integrated platform carries the potential to investigate the different secretion dynamics of cells, tissues, and spheroids in OOC platforms, and can be used in new applications, especially in drug screening and personalized medicine technologies. First, we presented here the development of a cellulose-based scaffold to embed pancreatic islets, which provide adequate mechanical properties to biomimic its native architecture. This approach could be extrapolated to other biological systems which require a soft, biocompatible, and non-biodegradable environment to biomimic physiological conditions. The scaffold is integrated into a versatile transparent microfluidic bioreactor to provide the medium exchange, simulate the physiological conditions, and optical monitoring of the islet morphologies. Finally, we developed an automated modular platform that used a microfluidics-controlling breadboard for the timed routing of fluids to interface with an LSPR biosensor chip for measuring soluble biomarkers, such as insulin in situ. All sensing was performed in situ in an uninterrupted and automated manner, allowing for the long-term monitoring of insulin secretion under external glucose stimuli for up to 3 h. We believe that our integrated modular on-line fluid routing and biosensing platform will be compatible with existing tissue organoid models and will promote their performance in drug screening by providing the capability for the real-time in situ monitoring of their microenvironment.

Even though the experimental set-ups we presented here are not optimally miniaturized, at the initial prototyping stage, such a platform has allowed us to validate our approach to biosensor integration. Combining, for the first time, these two unique technologies will open up new avenues of research into metabolic pathologies in a bid to meet the strong need for the combination of organ-on-a-chip system with microfluidics-integrated, non-invasive biosensing modules to achieve continual bioanalysis of microtissue behaviors. These themes are in line with current efforts to find new techniques to reduce the amount of animal testing, to provide personalized medicine, and to understand the onset and progression of diabetes.

## Figures and Tables

**Figure 1 biosensors-11-00138-f001:**
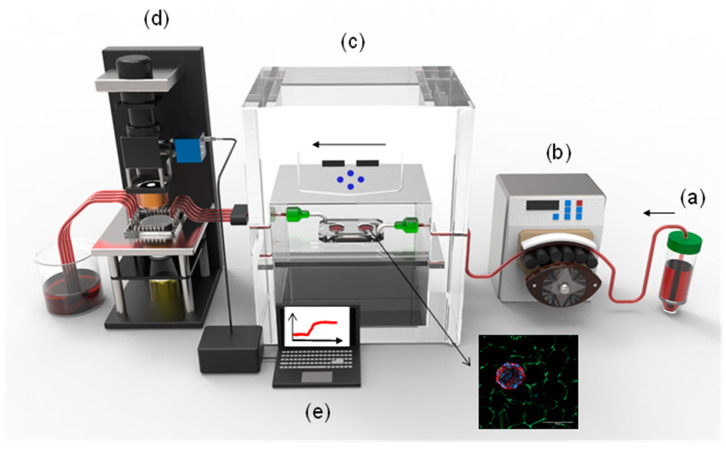
Schematic overview of the integration of the islet-on-a-chip (IOC) device with the on-chip LSPR sensing platform. (**a**) KRBH buffer with a chosen glucose content; (**b**) a peristaltic pump to drive the buffer into the IOC device; (**c**) IOC device containing mouse islets embedded in a cellulose-based scaffold; (**d**) the LSPR sensing platform to interrogate the buffer from the IOC device; and (**e**) monitoring of the insulin detection as a consequence of glucose stimulation.

**Figure 2 biosensors-11-00138-f002:**
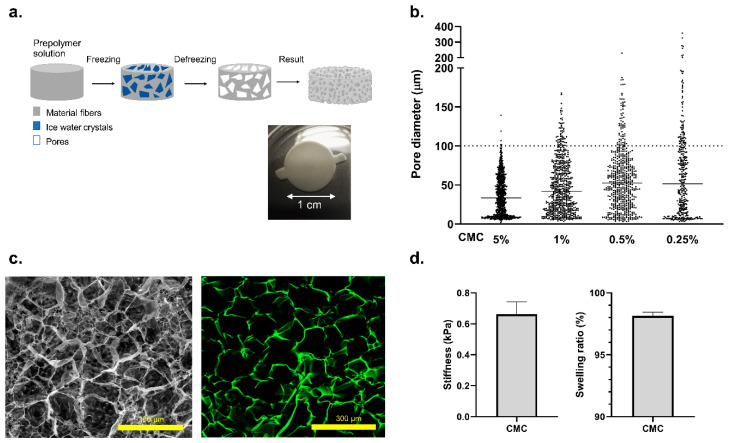
Cellulose-based cryogel fabrication and characterization. (**a**) General overview of the CMC cryogel fabrication protocol. Real images of CMC cryogel 0.5% (*w*/*v*) before islet seeding. Dimensions are 0.5 cm in height and 1 cm in diameter. (**b**) Pore diameter distribution of the cryogel at different CMC concentrations: 5%, 1%, 0.5%, and 0.25% (*w*/*v*), respectively. A total of 3 replicates and 20 images from 5 different depths were analyzed. (**c**) SEM image of the 0.5% (*w*/*v*) CMC cryogel condition after critical point drying. A confocal image of the same sample stained using 1 mM aminofluorescein (green) is on the right. (**d**) Characterization of the mechanical properties of the 0.5% (*w*/*v*) CMC cryogel. Young’s modulus of 0.67 ± 0.1 KPa was obtained by compression assays with 5 N load cell. Data corresponds with 3 compressions per cryogel and n = 3. The swelling ratio was determined obtaining values of 98.1% ± 0.3% for replicates n = 3 and 3 measures per cryogel. Values are expressed as mean ± SD: *p* < 0.05.

**Figure 3 biosensors-11-00138-f003:**
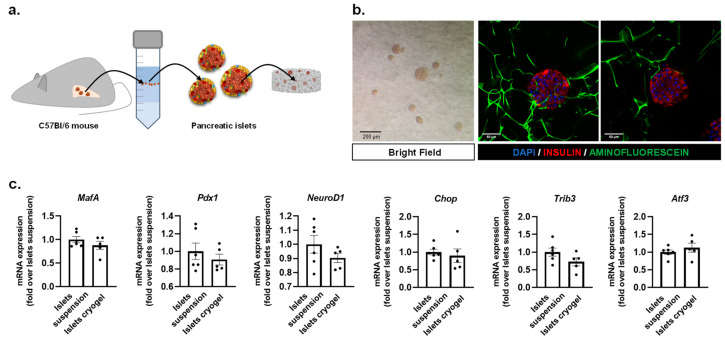
Development and characterization of the CMC islet construct. Gene expression analysis and immunostaining assays of islets were performed 24 h after being seeded within the cryogel. (**a**) Schematic diagram of mouse islet isolation. Islets inside the cryogel are also represented. (**b**) Left: pancreatic islets embedded within a carboxymethyl cellulose (CMC) cryogel under bright field (scale bar: 200 μm); middle and right: images of islets stained with insulin (red) and DAPI (blue). Cellulose fibers are stained with fluorescein (green). Images show islets at different depths through the cryogel (along the z-axis) (scale bar: 50 μm). (**c**) Gene expression analysis of *MafA*, *Pdx1*, *NeuroD1*, *Chop*, *Trib3*, and *Atf3* from islets in suspension and islets housed within the cryogel. Gene expression data were normalized against *Tbp1* and are shown relative to islets in suspension. Results are expressed as the mean ± SEM from three independent experiments. A *t*-test was applied to compare the data set, evidencing no statistical differences between islets in suspension and islets in the cryogel.

**Figure 4 biosensors-11-00138-f004:**
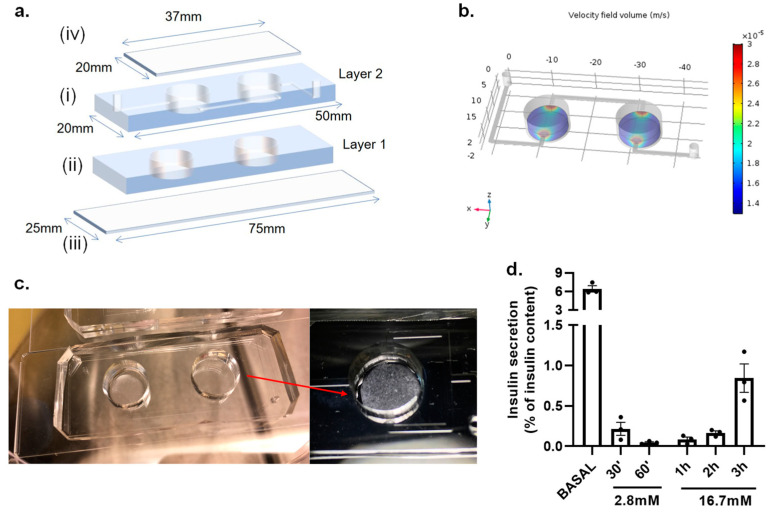
Design and fabrication of the IOC device. (**a**) Schematic image showing the assembling of the IOC: (**i**) a 3 mm layer of PDMS containing channels, inlet and outlet (Layer 2); (**ii**) a 2 mm layer of PDMS with the chambers for CMC islets (Layer 1); (**iii**) two-layers of microfluidic chip is bound on a (25 mm × 75 mm) standard cover slide; and, finally, (**iv**) a customized (37 mm × 20 mm) cover slide is used to sealed tissue chambers. (**b**) COMSOL Multiphysics^®^ simulation of the flow velocity and dynamics through the IOC device showing a maximum velocity of 0.03 mm s^−1^ (red zones) at the boundaries of the scaffold. (**c**) Real picture of the IOC device with a close-up view of the CMC islets fabricated inside the chamber. (**d**) A glucose-stimulated insulin secretion (GSIS) assay was performed in static conditions to evaluate the secretory capacity of pancreatic islets housed within the CMC-based cryogel inside the device. Results are expressed as mean ± SEM from three independent experiments.

**Figure 5 biosensors-11-00138-f005:**
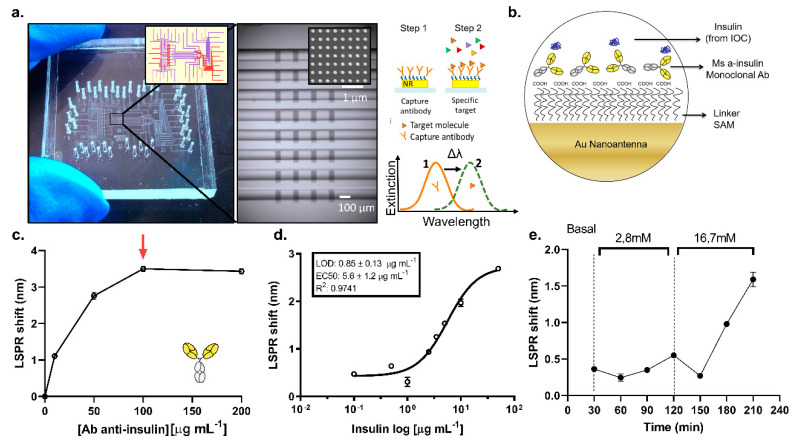
On-chip LSPR sensing platform description and detection. (**a**) Overview of LSPR sensing platform integrated by functionalized gold nanoantennas couple with a complex microfluidic network. (**b**) Functionalization strategy applied on gold nanoantenna sensors. (**c**) Optimization of capture monoclonal anti-insulin antibody. A saturation plateau is observed at 100 µg mL^−1^ antibody concentration (red arrow). (**d**) Calibration curve performed in KRBH basal glucose content (11.1 mM). Curve shows a limit of detection of (0.85 ± 0.13) µg mL^−1^ (n = 3). (**e**) Real-time insulin detection by the LSPR sensing platform every 30 min from the connected IOC device stimulated with KRBH buffer at low (2.8 mM) and high glucose (16.7 mM) concentrations, respectively.

**Table 1 biosensors-11-00138-t001:** Primer sequences used for gene expression analysis for qPCR.

Gene	Species	Fw	Rv
***MafA***	Mouse	CAAGGAGGAGGTCATCCGAC	TCTCCAGAATGTGCCGCTG
***Pdx1***	Mouse	CCCCAGTTTACAAGCTCGCT	CTCGGTTCCATTCGGGAAAGG
***NeuroD1***	Mouse	GGATCAATCTTCTCTTCCGGTG	TGCGAATGGCTATCGAAAGAC
***Ddit3/Chop***	Mouse	TCATCCCCAGGAAACGAAGAG	GCTTTGGGATGTGCGTGTG
***Trib3***	Mouse	CGTGGCACACTGCCACAAG	TCCAGGTTCTCCAGCACCAG
***Atf3***	Mouse	GTCCGGGCTCAGAATGGAC	CGTGCCACCTCTGCTTAGCT
***Tbp1***	Mouse	ACCCTTCACCAATGACTCCTATG	ATGATGACTGCAAATCGC

## Data Availability

The data presented in this study are available on request from the corresponding author. The raw/processed data required to reproduce these findings cannot be shared at this time as the data also forms part of an ongoing study.

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
