# Peer review of "In Situ LSPR Sensing of Secreted Insulin in Organ-on-Chip"

_biosensors, 2021, doi:10.3390/bios11050138_

Round 1

Reviewer 1 Report

The manuscript by Ortega et al. described an organ-on-a-chip platform integrated with an on-chip localized surface plasmon resonance (LSPR) sensing module capable of in-situ monitoring insulin secreted from the islets. An islet-on-a-chip (IOC), consisting of microfluidic PDMS channels with a pancreatic islet-laden carboxymethylcellulose (CMC) scaffold in each of the two incubating chambers, provides a dynamic environment in which the pancreatic islets can be cultured. The functionalized gold nanoantennas in the LSPR sensing module then captures the insulin from the pancreatic islets in the IOC, causing a shift in the LSPR which can then be monitored at a specific time.

Overall, the manuscript well outlines the function and capacity of each element in the proposed system and includes the main data to demonstrate the potential of the system. However, some additional explanations such as the design of the microfluidic devices, simulation modeling, and data analysis were required to improve the reliability of the data and interpretation.  

The following key points should be addressed for the manuscript to be suitable to be published in this journal, Biosensors.

Specific comments:

 1) The authors should summarize why real-time insulin monitoring in an IOC is necessary as this aspect is not shown by the performed experiments. Why do the insulin levels in an IOC have to be monitored continuously, and how can the monitored data be used to better treat type 2 diabetes? More importantly, the authors should note that the system cannot objectively be seen as “real-time” or “continuous” as it uses eight distinct channels to monitor insulin levels at discontinuous times and such terminology should be removed to avoid confusion.

2) The authors said that this integrated platform could be used in applications such as drug screening platform for human models surpassing animal testing. In this experiment, mouse pancreatic islets were used, not human’s. Then, how do the authors plan to develop the model with which samples? Will the human primary pancreatic islet be used?

3) Primary mouse pancreatic islets were embedded in a non-biodegradable cellulose-based scaffold. Although the cellulose-based scaffold is robust, it is basically different from the extracellular matrix in biological characteristics. An explanation of the advantages other than the large pore diameter and the porosity of this scaffold in the in vitro system is needed.

4) In section 2.2, “Characterization of CMC cryogels,” the cryogel was submerged into MiliQ water for 4 days for equilibrium. Why were 4 days spent and how did the authors confirm that the cryogel was fully equilibrated?

5) The 0.5% CMC you used had a wide range of pore sizes from sub-micrometer to about 150 micrometers. When injecting islets into the cryogel, it can be difficult to fill the islets homogeneously into the CMC because of clogging issues in the pores smaller than islets. Are there any such problems? If so, how did you fill the islets homogeneously? In addition, the distribution of the islets within the scaffold should be analyzed. The spatial organization of the islets within the scaffold should be confirmed to show the extent to which the system can “biomimic the native pancreas host.”

6) In Figure 2a caption, the authors said that the cryogel was in dimensions of 0.5 cm in height and 1 mm of diameter. It seems typo error of 1 cm of diameter based on the text and photographs. However, it does not seem that the diameter is 1 cm in the left photograph of cryogel or the ratio is right. Also, there is no scale bar. What does the right photograph indicate? A protrusion is observed.

7) In the COMSOL simulation, a solid cylinder construct was designed in the chamber to replicate the flow characteristics inside the CMC scaffold. How were the material properties of the construct assigned to represent the CMC scaffold? In addition, it would be more appropriate to set the construct as a “porous medium” rather than a solid to analyze the flow characteristic inside the CMC scaffold. If it was a simulation model in which major conditions such as construct porosity were neglected, it is difficult to accept the author's claim on the stability of the 3D construct and the shear stress because the model cannot represent the CMC scaffold.

8) In Figure 3c, when the gene expression was analyzed after islet seeding in CMC-scaffold? β-cell-specific transcriptional regulators and positive indicators of β-cell health and functionality (MafA, Pdx1, NeuroD1) showed no statistically significant difference in mRNA expression between two samples, but islets embedded in cryogel showed lower expression on average. Doesn't the expression value fall further if the islets were left in the cryogel scaffold for a longer time?

9) If there are 30 islets in a 1 cm diameter, 5 mm height cryogel in this experiment, is the density of the islets similar to that of in vivo?

10) The right panel in Figure 3b shows islets in different depths of the cryogel. Then, the depth should be indicated.

11) In the device developed for the experiment, two chambers for loading CMC were connected by a fluidic channel. In addition, an inlet and outlet channel were connected to each chamber for introducing fluid. The fluid injected through the inlet port passes through the two chambers (two CMCs) sequentially and exits through the outlet channel. Despite not having studied the interaction between the CMCs of each chamber, why did you design the two chambers in the device?

12) There is no Figure S1 in the supporting files.

13) In Figure 4d, basal secreted 6% of insulin in 30 min. CMC-islets secreted less than 1% of insulin even after 3 h. Of course, the secretion indeed increased over time, but is it significant?

14) More explanation about LSPR sensing and sensing platform is needed. Figure 5a is not easy to understand in terms of device design. In addition, in Figure S2b, 200 μg/mL of antibody seems to shift more than 100 μg/mL, but not in Figure 5c.

Reviewer 2 Report

This is a well-conducted study with the right controls being done. The data looks good and provides good support for their assertion that their system is working well. The data are well presented, making it easy to follow their arguments. Excellent images are presented which further aids the readers in understanding the work done. The manuscript is also well-written, enabling the reader to understand the essentiality of the project, the experimental set-up, the results, the conclusions, and more importantly, where this research is going next. The only way I can suggest to improve the manuscript is to include more relevant references pertaining to microfluidic systems for organ-on-chip applications. This is an example:

Yu, F., Iliescu, F.S. and Iliescu, C., 2016. A comprehensive review on perfusion cell culture systems. Informacije MIDEM46(4), pp.163-175.

Author Response

Reviewer 2:

This is a well-conducted study with the right controls being done. The data looks good and provides good support for their assertion that their system is working well. The data are well presented, making it easy to follow their arguments. Excellent images are presented which further aids the readers in understanding the work done. The manuscript is also well-written, enabling the reader to understand the essentiality of the project, the experimental set-up, the results, the conclusions, and more importantly, where this research is going next. The only way I can suggest to improve the manuscript is to include more relevant references pertaining to microfluidic systems for organ-on-chip applications. This is an example:

Yu, F., Iliescu, F.S. and Iliescu, C., 2016. A comprehensive review on perfusion cell culture systems. Informacije MIDEM46(4), pp.163-175.

We highly appreciate the reviewer’s comments regarding the study presented in this manuscript. As suggested by the reviewer we have incorporated more relevant references to support our work, starting from the one you suggested.

Round 2

Reviewer 1 Report

Thank you for the efforts to revise the manuscript by Ortega et al.  Compared to the previous version, the revised manuscript improved the quality and presentation of the work.
Thus, I would recommend for publication in Biosensors.